# Informally Vended Sachet Water: Handling Practices and Microbial Water Quality

**David Manjaya [1], Elizabeth Tilley [2,3,*] and Sara J. Marks [3]**

1   College of Medicine, University of Malawi, Private Bag 360, Chichiri, Blantyre 3, Malawi; manjayad@gmail.com
2   Department of Environmental Health, University of Malawi, The Polytechnic, Private Bag 303, Chichiri, Blantyre 3, Malawi
3   Department of Sanitation, Water, and Solid Waste for Development, Eawag: Swiss Federal Institute of Aquatic Science and Technology, Ueberlandstrasse 133, 8600 Duebendorf, Switzerland; sara.marks@eawag.ch
*   Correspondence: elizabeth.tilley@eawag.ch

**Abstract:** Informally vended water is an important source of water for marginalized people who do not have access to formal or public sources. In Malawi, hand-tied sachets of water are common but not regulated, and the quality of the water and hygienic practices during packaging are unclear. We analyzed microbial concentrations in the source water (origin), internal water (packaged) and on the external surface (plastic bag) of sachets from 76 vendors operating in the busy Mwanza crossing into Malawi from Mozambique. The results indicated that the majority (75%) of the water sources met the WHO guidelines (<1 CFU/100 mL) for potable water, while only 38% of the water inside packages met this guideline, indicating a sharp increase in contamination due to packaging and handling practices. The external surface was highly contaminated and is the point of contact between the consumer's mouth and the liquid within; furthermore, external contamination was a strong and significant predictor of internal contamination. We advise against strict enforcement that would limit access to this important drinking water source, but recommend hygiene education for vendors that focuses on filling and storage, refrigeration and especially ensuring sanitary coolers from which the bags are sold in order to limit re-contamination during handling.

**Keywords:** drinking water; informal business; sustainable development goals; Malawi; Africa; access; hygiene

## 1. Introduction

The consumption of packaged water in high-income countries has increased rapidly in recent decades, with low- and middle-income countries following suit more recently [1–4]. The WHO/UNICEF Joint Monitoring Program that monitors progress towards the Sustainable Development Goals (SDGs) classifies packaged water as an "improved source" despite the fact that national standards vary wildly, if they exist at all [5]. Drinking water may be packaged in plastic, glass bottles or sachets [1] and may be produced by formal (registered) bodies or informal (unregistered) individuals with varying qualities of the source water [6]. Due to the informal nature of the industry, documentation of the extent, quality or practices of unregistered water vending businesses is limited despite being common across Africa [4,7].

Sachet water is typically a 500 mL polyethylene plastic bag of water that can be sealed mechanically by heat sealing all four sides or by simply tying a knot; the former is produced by the formal sector, while the latter are popular among informal entrepreneurial water vendors [2]. However, studies conducted in West Africa reported that sachet drinking water generally does not meet safe standards for drinking water [1,8,9]. In Nigeria, for example, 30% of sachet water samples had fecal coliforms

above the recommended WHO guideline of non-detectable *Escherichia coli* per 100 mL [6,10]. In their systematic meta-analysis of 57 studies in 30 developing countries, Wright et al. reported that all water collected was exposed to bacterial contamination during transport and storage implying that source quality is not a sufficient predictor of water consumed [11]. Water vendors may additionally contaminate the water while hand-filling and hand-tying the sachets [12]. Furthermore, when filling, vendors may blow air into the bags to force them open and thus increase the level of microbial contamination. Even factory-bottled water can be highly contaminated in unregulated contexts [13]. For this reason, WHO recommends routine risk assessments of drinking water supplies intended for vending, including the monitoring of the source water quality, distribution methods and the design and characteristics of containers used to collect, transport and store the water [6].

Due to its affordability, convenience and transportability, sachet water is crucial for many households in low- and middle-income regions; however, the overall desirability of sachet water from a public health and urban planning perspective remains uncertain [14]. For example, concerns have been raised about the role of sachet water in exacerbating outbreaks of cholera and other waterborne diseases in Nigeria [10]. This concurs with a case-control study conducted in Sierra Leone, which revealed that consuming vended drinking water of any type was a significant risk factor for cholera infection [15]. In addition, more than 80% of diarrhea patients from a study in Accra, Ghana reported consuming sachet water (though numerous confounding factors were reported) [16]. Yet contrary to these findings, in water-scarce environments, vended water serves an important role: One study in Ghana found that sachet water use was associated with higher levels of overall health in women and reduced the likelihood of diarrhea in children [2].

In Malawi, the local government regulates the water-packaging sector, and several companies package drinking water in plastic bottles, which are sold for 250 MK/500 mL on average. Hand-filled sachet drinking water is also locally available and cheaper than factory-bottled water at an average price of 50 MK/500 mL (50 Malawian Kwacha (MK) = 0.07 USD). Unlike bottled water, which is usually sold in shops and kept in refrigerators, sachet water is mostly sold on the streets from cooler boxes. Though sachet water falls within the scope of packaged water regulation, the informal sector appears to be largely, if not wholly unregulated.

Informally packaged water also plays an important role in humanitarian settings when traditional water sources become contaminated, inaccessible or disappear [17]. Cyclone Idai hit the eastern coast of Mozambique on 14 March, 2019 and continued to leave destruction across Malawi and into Zimbabwe. After only two weeks, over 700 people had been confirmed dead with many more expected, thousands were displaced and much of Mozambique's port city of Beira was underwater; the first cases of cholera were confirmed [18]. It is precisely within these types of dire situations that informally vended water markets emerge to meet the needs of transient, desperate consumers with no or few other options.

At the time of publication, the authors know of no study on the quality or management of informally vended sachet water in Malawi. The objectives of this study were to assess the handling practices of water-sachet vendors in the context of an urban center in Malawi, to identify the potential causes of contamination and to determine whether the water met international guidelines for drinking water.

## 2. Materials and Methods

### 2.1. Study Site

This research took place in Mwanza, Malawi, which is in the south-west corner of the country. Mwanza has a population of about 10,000 and is the administrative headquarters for Mwanza district. Tete, on the Mozambican side, is a large, industrial city, through which many goods destined for Malawi pass. As such, a considerable service industry that caters to the travelers and business people crossing the border has emerged in Mwanza; packaged water is widely available and in high demand.

## 2.2. Ethical Approval

Written consent was sought before an individual was included into the study, including consent to collect water samples for analysis in the laboratory. There were no refusals to participation from the subjects approached. The study was approved by the University of Malawi College of Medicine Research and Ethics Committee (COMREC) approval number P.10/16/2033.

## 2.3. Sampling Procedure

In this quantitative cross-sectional study, all water-sachet vendors ($N = 76$) who were working at one of the three busy markets in Mwanza were interviewed; vendors working from their homes or at other non-public locations were not included in the sample. All water vendors who sold water directly from a jerry can (20 L plastic jug) or factory-bottled water were also excluded from the study. Sachet water vendors were interviewed at the place of sale to obtain data on processing and handling of the water samples. The questionnaires were delivered orally in the local language (Chichewa) in order to understand water handling practices.

Following the interview, a bag of sachet water was purchased to be used in water sampling. First, the sachet was placed into a bag containing 100 mL of distilled water and shaken for a minute in order to collect an exterior sample (distilled water was produced at the district hospital). This method was adapted from Fisher et al. who rinsed with 300 mL; although exterior surface samples may be reported as a function of area, we instead made use of the rinse-based methods developed in the hand hygiene literature (e.g., see the work by Pickering et al. [19,20]) using 100 mL of rinse water to standardize units and prevent excessive dilution [1]. After shaking, both the exterior sample and the sachet were immediately placed in a cooler box containing ice packs. Later, either with the vendor or with information from the vendor, samples of the water source used were also collected. The source water samples were collected by filling a plastic bottle with water directly from the source outlet after letting the water run for approximately 10 s; the outlet rim was not flamed or disinfected. The three samples were then transported in a cooler box with ice packs to a laboratory at the District Hospital in Mwanza for processing. In total, 222 water samples were collected, of which 76 samples were interior point-of-sale, 76 were exterior point-of-sale and 70 were source water samples (because of multiple vendors using the same source) (Table 1).

**Table 1.** Description of water sampling at three locations.

| Sample Location | Sample Size | Description |
|---|---|---|
| Source water | 70 | Source of sachet water as provided by the vendor |
| Point of sale exterior | 76 | Sachet exterior surface was rinsed in sterile water after purchasing |
| Point of sale interior | 76 | Contents of the sachet |

## 2.4. Laboratory Analyses

Upon arrival at the lab, the water samples were immediately refrigerated and analyzed within 6 h. The samples were analyzed for total coliforms and *Escherichia coli* (*E. coli*) using the membrane filtration method. The water sachets were aseptically opened using a flame-treated pair of scissors. A 100 mL water sample was filtered through a 0.45 μm Millipore membrane using a DelAgua filter device (www.delagua.org). The filters were then placed on Hyserve Compact Dry Plates (https://hyserve.com) and incubated at 37 °C for 24 h. Thereafter, any colonies formed were counted in colony forming units (CFU) per 100 mL of the water sample. Additionally, samples were processed using only 1 mL of sample (pipetted directly onto the Dry Plate and processed as above) to allow for enumeration of colonies among exceptionally contaminated samples. All sampling points were expressed in units of CFU/100 mL for comparison purposes. Since each sachet bag was rinsed in 100 mL of distilled water, the expressed concentration for exterior samples was also equal to the total CFU count per sachet.

*2.5. Quality Control*

Quality control involved the daily preparation of blanks (negative controls) and positive controls. Blanks were prepared using distilled water and positive controls were prepared by mixing a pea-sized amount of chicken feces with water. Both of the positive and negative controls were then analyzed according to the water testing protocol. Furthermore, every tenth sample was analyzed in duplicate.

*2.6. Data Analyses*

Questionnaires were checked for errors and missing data. Data were entered manually into Excel where it was cleaned before it was exported to Stata version 12 for statistical analysis (StataCorp, College Station, TX, USA).

## 3. Results

*3.1. Demographics*

The results of the 76 questionnaires are summarized in Table 2. We determined that the water vendors were generally young women with a high level of education (71% completed secondary school or higher). The water vendors do most of the water fetching and packaging themselves, with little, if any assistance. The water, however, is time consuming to obtain, requires more than one trip daily to collect and is not always available. It is not clear from our results if the source water collected was from the vendors' reported primary source or an alternative source, which is a shortcoming of this work.

All vendors paid for their water (even the person who reported using a shallow well) and a majority of respondents obtained their water from a piped source (59%). The water was usually stored with a lid and removed with a cup, but only 27% of vendors reported treating their water; if it was treated, chlorine was the most common method (18%).

Bags were sealed by hand (i.e., knots are tied), with a small number of vendors blowing into the bag to inflate it before filling. Nearly all sachets were refrigerated at home before selling and were kept cold in a cooler during the day, though only about a third of vendors actually kept ice in the cooler, instead relying on the cooler to keep the pre-chilled sachets cold. Electricity is rare, intermittent and expensive in Malawi, which may explain the higher level of education among the vendors, i.e., the higher education levels are a proxy for income and therefore access to the electricity and refrigeration necessary to succeed in the water-vending business.

There was no price competition: All vendors sell their water at 20 MK (about 0.03 USD) and sell about 80 bags a day, though the range was large and likely varied due to their location within the markets and the proximity of competition.

**Table 2.** Summary of water vendor characteristics.

| Variable | | Sample Size | Value |
|---|---|:---:|:---:|
| Male | | 76 | 36% |
| Mean (SD) age | | 75 | 29 (10) |
| Education level completed | | 76 | |
| | None | | 4% |
| | Primary | | 25% |
| | Secondary | | 62% |
| | Tertiary | | 9% |
| Responsible for | | 76 | |
| | Fetching water | | 87% |
| | Treating water | | 13% |
| | Packaging water | | 80% |

**Table 2.** *Cont.*

| Variable | | Sample Size | Value |
|---|---|---|---|
| Water source | | 76 | |
| | Piped to house | | 13% |
| | Piped to yard | | 42% |
| | Piped to neighbor | | 4% |
| | Public tap | | 3% |
| | Public borehole | | 37% |
| | Shallow well | | 1% |
| Mean (SD) time spent to collect (minutes round trip) | | 47 | 69 (58) |
| Mean (SD) number of trips per day | | 55 | 1.8 (0.8) |
| Water is stored with a lid | | 76 | 80% |
| Water is used directly without storage | | 76 | 9% |
| Water is removed with a cup (not tap) | | 76 | 76% |
| Mean (SD) price per 20 L of water (MK) | | 76 | 200 (0) |
| Water is treated | | 76 | |
| | Boiling | | 3% |
| | Chlorine | | 18% |
| | Cloth | | 3% |
| | Sedimentation | | 3% |
| | No treatment | | 70% |
| | Do not know | | 3% |
| Mean (SD) time spent packaging (minutes) | | 71 | 55 (22) |
| Mean (SD) sachets packaged per day | | 76 | 92 (55) |
| Water is packaged manually | | 76 | 100% |
| Sachets are open by blowing | | 67 | 3% |
| Bags are sealed by tying | | 69 | 100% |
| Sachets are refrigerated at home | | 73 | 99% |
| Sachets are sold from a cooler | | 71 | 99% |
| Coolers contain ice | | 74 | 34% |
| Mean (SD) price per sachet (MK) | | 76 | 20 (0) |
| Mean (SD) sachets sold per day | | 76 | 81 (52) |

*3.2. Microbiological Water Quality*

Based on *E. coli* measurements, the WHO defines four levels of risk for drinking water [6]:

- Conformity: <1 CFU/100 mL
- Low: 1–10 CFU/100 mL
- Intermediate: 11–100 CFU/100 mL; and
- High: >100 CFU/100 mL

The results for the analyses of the 222 samples were categorized according to these guidelines and the results are shown for *E. coli* (left) and total coliforms (right) in Figure 1. Though total coliforms are not useful as an indicator of fecal contamination, they are useful to assess the cleanliness and disinfection efficacy. The results for both indicators showed that generally, the majority of source water samples met the criteria for the conformity or low risk classification, though a non-trivial number of samples fell beyond this threshold. Contamination increases as the water was packaged: Internal water samples showed higher levels of both *E. coli* and total coliform, and lower levels of samples reaching the conformity standard. External samples had the highest level of contamination and the lowest share of samples with undetectable indicator bacteria. The distribution of the measured concentrations across risk categories are summarized in Table 3.

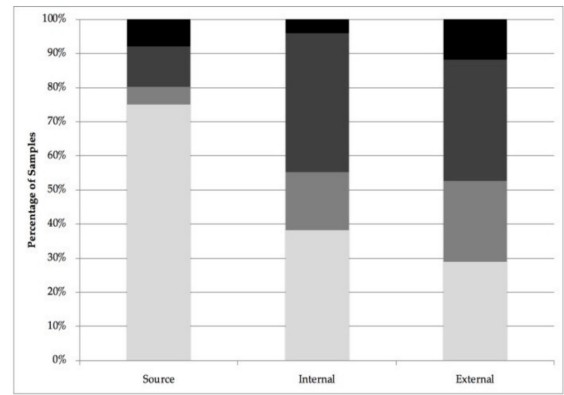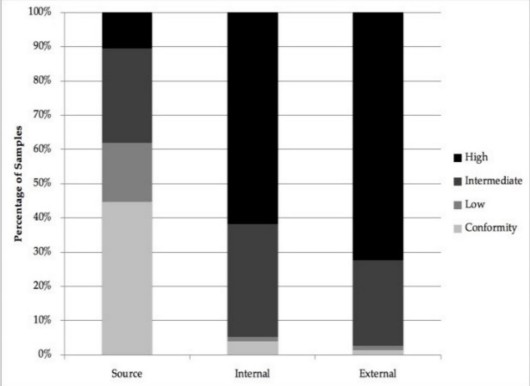

**Figure 1.** Contamination levels for *Escherichia coli* (**left**) and total coliforms (**right**).

**Table 3.** Summary of the microbiological results (% samples in each category and (*N*)). Geometric mean concentrations are in units of CFU/100 mL.

|  | Source Water | | | | Internal | | | | External | | | |
|---|---|---|---|---|---|---|---|---|---|---|---|---|
|  | *E. coli* | | TC | | *E. coli* | | TC | | *E. coli* | | TC | |
| Conformity * | 75% | (57) | 45% | (34) | 38% | (29) | 4% | (3) | 29% | (22) | 1% | (1) |
| Low | 5% | (4) | 17% | (13) | 17% | (13) | 1% | (1) | 24% | (18) | 1% | (1) |
| Intermediate | 12% | (9) | 28% | (21) | 41% | (31) | 33% | (25) | 36% | (27) | 25% | (19) |
| High | 8% | (6) | 11% | (8) | 4% | (3) | 62% | (47) | 12% | (9) | 72% | (55) |
| Geometric mean | 1.32 | | 1.76 | | 1.86 | | 3.41 | | 3.62 | | 2.05 | |
| [95% Conf. Interval] | [1.18–1.48] | | [1.56–2.00] | | [1.65–2.10] | | [3.19–3.65] | | [3.45–3.80] | | [1.83–2.30] | |

* Risk categories are defined according to [6] where conformity = <1 CFU/100 mL, low = 1–10 CFU/100 mL, intermediate = 11–100 CFU/100 mL, high = >100 CFU/100 mL.

### 3.3. Bivariate Analysis

Bivariate comparisons of fecal indicator concentrations at different sample locations indicated significantly lower levels of total coliform and *E. coli* at the source as compared to interior or exterior samples (Table 4). However, the Mann-Whitney U test showed no significant difference between *E. coli* concentrations of exterior (median (Mdn) = 9) and interior (Mdn = 6) samples ($U = 2502$, $p = 0.149$, $r = 0.117$). Similarly, the correlation analysis indicated that bacterial concentrations at the source were not correlated with those at the other two sampling points. However, interior and exterior contamination levels were moderately correlated for both *E. coli* ($r_s = 0.46$, $p < 0.001$) and total coliform ($r_s = 0.33$, $p = 0.004$).

**Table 4.** Bivariate comparisons of *E. coli* (EC) and total coliform (TC) concentrations (in CFU/100 mL) by point of collection using Wilcoxon signed rank (related samples), Mann-Whitney U (unpaired data) and Spearman's rank-order correlation tests.

|  | Statistic | | Source | | Interior | | Exterior | |
|---|---|---|---|---|---|---|---|---|
|  |  |  | EC | TC | EC | TC | EC | TC |
|  |  |  | *N* = 70 | *N* = 70 | *N* = 76 | *N* = 76 | *N* = 76 | *N* = 75 |
|  | Mean |  | 3 | 16 | 24 | 1000 | 44 | 1562 |
|  | (SD) |  | (7) | (28) | (38) | (2658) | (98) | (3240) |
|  | Median |  | 0.5 | 2 | 6 | 145 | 9 | 176 |
|  | Wilcoxon | Z |  |  | 4.75 | 7.01 | 5.16 | 7.19 |
|  |  | *p* |  |  | <0.001 | <0.001 | <0.001 | <0.001 |
|  |  | U |  |  | 1378 | 442 | 1115 | 243 |
| Source | Mann Whitney | *p* |  |  | <0.001 | <0.001 | <0.001 | <0.001 |
|  |  | *r* |  |  | 0.461 | 0.728 | 0.543 | 0.791 |
|  | Spearman rank correlation | $r_s$ |  |  | 0.038 | 0.184 | −0.115 | 0.115 |
|  |  | *p* |  |  | 0.754 | 0.127 | 0.345 | 0.348 |

**Table 4.** *Cont.*

| | Statistic | | Source | Interior | Exterior | |
|---|---|---|---|---|---|---|
| Interior | Wilcoxon | $Z$ | | | 2.28 | 2.26 |
| | | $p$ | | | 0.023 | 0.024 |
| | Mann Whitney | $U$ | | | 2502 | 2313 |
| | | $p$ | | | 0.149 | 0.046 |
| | | $r$ | | | 0.117 | 0.163 |
| | Spearman's rank-order correlation | $r_s$ | | | 0.460 | 0.333 |
| | | $p$ | | | <0.001 | 0.004 |

*3.4. Regression Analysis*

To estimate the potential factors contributing to the contamination level in the internal water, relevant variables were regressed onto the total coliform and *E. coli* concentration and log-transformed values. The results are presented in Table 5. All variables from the vendor questionnaire that were considered to impact water quality were included.

**Table 5.** Regression results.

| Variable | $\log(TC_{internal})$ | $TC_{internal}$ | $\log(EC_{internal})$ | $EC_{internal}$ |
|---|---|---|---|---|
| $\log(TC_{source\ water})$ | 0.251 | | | |
| $\log(TC_{external})$ | 0.100 | | | |
| $TC_{source}$ | | −3.600 | | |
| $TC_{external}$ | | 0.029 | | |
| $\log(EC_{source})$ | | | 0.084 | |
| $\log(EC_{external})$ | | | 0.520 *** | |
| $EC_{source}$ | | | | −0.237 |
| $EC_{external}$ | | | | 0.137 *** |
| Gender | 0.272 | 588.816 | −0.489 | 2.365 |
| Does fetching | 0.797 | 3550 * | −0.608 | −18.784 |
| Does treating | 0.682 * | 2620 ** | 0.796 | 15.343 |
| Does packaging | −0.573 | −1354.672 | 1.574 * | 39.315 |
| Improved source | 0.128 | 140.975 | −0.972 ** | −11.316 |
| Water covered | 0.208 | 933.379 | 0.111 | −1.483 |
| Cup for water bucket | −0.095 | −1283.057 | −1.306 ** | −28.879 ** |
| Water treated | −0.362 | −1566.211 | −1.122 ** | −26.844 ** |
| Ice in cooler box | −0.295 | −169.121 | 0.481 | 12.861 |
| Constant | 2.213 ** | 3415.391 | 3.213 ** | 44.026 |
| $N$ | 64 | 64 | 65 | 65 |
| R-squared | 0.228 | 0.267 | 0.445 | 0.338 |
| Adjusted R-squared | −0.013 | 0.038 | 0.276 | 0.135 |

* denotes significance at the 10% level, ** denotes significance at the 5% level, *** denotes significance at the 1% level.

The four models were tested for heteroskedasticity using the Breusch-Pagan tests. We cannot reject the null hypothesis that the error variances were equal in models 1 and 2 ($p = 0.112$ and $0.610$, respectively), though we cannot make this claim for models 3 and 4 ($p = 0.000$ and $0.064$, respectively).

We found strong and significant impacts from the external *E. coli* contamination on the internal *E. coli* contamination, though there was no similar relationship observed in the total coliform models. This finding may seem counter-intuitive in that external contamination occurs after the water has been packaged, and therefore cannot contaminate the internal water; however, in this case, the external contamination is a proxy for general handling hygiene of the packaging process. External contamination is the result of contaminated hands and surfaces, which we measured on the final product as we were not able to measure hand and surface contamination through the process. Additionally, we found no impact from the source water on the internal concentration for either bacteria type. These findings suggest that either there was little or no growth from the low-levels of contamination that were

measured in the source that bacterial die off may have negated any growth that did occur, or that net growth was overwhelmed by contamination during processing. In any case, the quality of the source was far less important to the interior water quality than other handling and packaging behaviors.

"Does treating", "Does fetching" and "Does packaging" were binary variables indicating whether or not the vendor does this activity her/himself; it was included to understand whether more or less hygienic practices were associated with the business owner or if contamination was introduced by employees. Unusually, we find an increased likelihood of contamination (columns 1 and 2) when the vendor does the treating her/himself, though treatment was rare, and this result was likely affected by "yay-saying" (i.e., untruthful answers to appease the interviewer).

"Improved source" was a binary variable constructed by grouping water sources that were labelled as "piped to house", "piped to yard", "piped to neighbor" and "public tap" as being "improved", while sources labelled as "public borehole" and "shallow well" were not. We found that using a treated source lowers the *E. coli* count by nearly 1 log unit (column 3) though there was no similar effect for total coliform.

The models showed that water treatment and the use of a cup for filling the bags produced strong and significant reductions in the *E. coli* values (columns 3 and 4), though no similar effect was observed for the total coliform models. This unexpected result (i.e., using a tap would be expected to be associated with a reduction in contamination) may be explained by the fact that water buckets that have taps are usually more robust and heavier than simple plastic buckets and are therefore less likely to be moved and washed; indeed, water can never fully be emptied so it becomes stagnant in the bottom and tap mechanisms cannot be dismantled and cleaned easily either. Though the use of a cup could introduce additional contamination, it is instead more likely that the use of a non-washable/non-washed tap-based bucket was more of a problem.

Additional models are provided in the Supplementary Materials (Table S1) to show model results that excluded the external water quality results as a covariate. The external results could on one hand be considered endogenous to the model as they may be correlated with the internal water quality, but on the other hand, they can also be viewed as proxies for hand hygiene, which may or may not affect the quality of the internal water, depending on how the water was packaged.

*3.5. Quality Control*

No total coliform or *E. coli* were detected from any of the 12 blank samples (negative controls). All 12 positive control samples contained high concentrations of total coliform and/or *E. coli* and most of the *E. coli* controls were too numerous to count. In terms of total coliform, 10 of the 12 duplicate samples contained the same bacterial concentration while two duplicate samples differed by 1–3 CFU/100 mL. Similarly, in terms of *E. coli*, 10 of the 12 duplicate samples had the same concentration while the remaining duplicate samples differed by 8–12 CFU/100 mL.

## 4. Discussion

Improved drinking water sources are considered to pose a lower risk to health than unimproved sources [21]. However, such sources, despite structural protections, do not guarantee water that is free from contamination (and vice versa). This study found that despite relatively clean water at the source level, high levels of fecal contamination were introduced via transport and handling processes by the time packaged water is available for purchase.

In Mwanza, piped water comes from surface water which is treated by the Southern Region Water Board before distribution. Intermittent operation is known to result in bacterial contamination through back-flow and infiltration as pressure is lost in the system [22]. So although the water is treated and managed by a competent authority, contamination is likely induced through distribution; vendors have no way to know or determine which of the supplies is best or most safe.

Additionally, the intermittent supply of piped water and dispersed alternative sources, means that water storage is a necessity in the water-vending business. It has been well documented that

post-collection contamination is a significant contributor to water contamination, regardless of how well it was covered or accessed [23–27].

Our results showed that the majority of samples taken from a sachet did not meet the conformity standard for *E. coli* and many contained levels that would place them in the highest risk category [6]. International drinking water quality guidelines stipulate that drinking water should have no detectable *E. coli* in any 100 mL sample. These findings were consistent with recent studies from Nigeria and Sierra Leone, which documented that 30% and 37% of sachet water samples were contaminated with *E. coli*, respectively [1,10].

The volume of water consumed, and therefore the risk associated with it, is greatest from the internal water. However, the bag's external contamination is also relevant as it is the point of contact between the consumer and the water stored within the sachet. In other words, the external contamination points to the processing as the cause of the internal contamination. External contamination of sachet bags also provides a basic indication of possible health risks due to contact during consumption. The consumer places his/her mouth over the corner of the sachet, rips off a piece of the plastic and drains the liquid by sucking it out. In the process, his/her mouth is in direct contact with the external surface and, as he/she drinks, is passing the internal liquid over the external surface, thereby transferring additional contamination into the liquid consumed. We note that the literature reports varying recovery rates for *E. coli* when using soak- or rinse-based methods, ranging from 52% for hands [20] to 60% or greater for plastic surfaces [28,29]. Therefore, the external contamination levels reported here are likely a conservative estimate of the true value.

Importantly, majority of the source water samples had relatively low-levels of bacterial contamination and would be categorized at being in the conformity or low risk groups. We found no relationship between the contamination of the source water and internal sachet contamination. Furthermore, external contamination levels were both higher than the source and there was a strong and significant association between the external and internal water quality (for *E. coli*). Therefore, hygiene and handling appear to play an important role in the quality of the sachet water: More so than the source.

Most vendors (70%) in this study did not treat their water before packaging it, which is consistent with other findings from Africa, where access to improved water sources was found to reduce the probability that people will treat their water at home [30,31]. Most vendors refrigerate their sachets, which could be responsible for limiting pathogen growth in the internal water samples, though it is unclear whether this practice reflects an understanding of good hygiene or simply the need to meet consumer demand for cold water in a very hot climate.

## 5. Conclusions

Sachet water is an important, affordable and easily accessible source of water for people in low-income countries, especially those that are living away from home by choice or because of a humanitarian crisis. Generally, the quality of water sold in Mwanza would not be considered acceptable for sale by a formal water vendor, despite the fact that the majority of water sources used were of an acceptable quality.

Realistically, the local government in Mwanza does not have the human or financial resources to monitor and/or enforce quality standards on the informal water sector, which would be the best scenario. Instead, short training and educational materials should focus on the use of improved water sources, frequent hand-washing with soap during the packaging process, the continued use of refrigeration prior to vending, the use of cold-packs during the day and perhaps most importantly, the frequent cleaning of the cool-boxes with a weak chlorine solution (e.g., one cap full of local bleach in 1 L water) to avoid external contamination to limit the quantity of pathogens that come directly in contact with the consumer's mouth. Similarly, consumers should be encouraged to pour the water into their mouths from a cut corner in order to minimize contact with the package exterior.

**Supplementary Materials:** The following are available online at http://www.mdpi.com/2073-4441/11/4/800/s1, Table S1: Model results excluding external water quality.

**Author Contributions:** Conceptualization, E.T. and S.J.M.; methodology, S.J.M; formal analysis, D.M., E.T. and S.J.M.; investigation, D.M.; resources, S.J.M.; data curation, E.T.; writing—original draft preparation, D.M., E.T. and S.J.M; writing—review and editing, E.T. and S.J.M.; supervision, E.T. and S.J.M.; project administration, E.T. and S.J.M.

**Funding:** This research was supported by The Swiss Federal Institute of Aquatic Science and Technology (Eawag) and by the UK Aid from the Department of International Development (DFID) as part of the Sanitation and Hygiene Applied Research for Equity (SHARE) Research Consortium (http://www.shareresearch.org). However, the views expressed do not necessarily reflect the Department's official policies.

**Acknowledgments:** The authors would like to thank the participating water vendors and the Mwanza District Hospital for letting us use their laboratory facilities.

**Conflicts of Interest:** The authors declare no conflict of interest

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
