# Peer review of "Informally Vended Sachet Water: Handling Practices and Microbial Water Quality"

_water, doi:10.3390/w11040800_

Round 1

Reviewer 1 Report

This is a very interesting paper on water quality in sachet water the would be of interest to readers of the journal. It is well written and well cited. There are a few potential problems with the statistical analysis.

Specific comments:

Figure 1 does not seem necessary as the study took place in one location.

Please provide company information (city or URL) for named commercial products.

Data analysis section lacks detail, including detail on the regression method. Any regression diagnostics used?

Consider moving Section 2.5 Data Analysis after Section 2.6 QAQC

Table 3 – what are the units of concentration?

Table 4 seems unnecessary; results can just be described.

What is the correlation between the contamination levels at the three sites?

Were paired or unpaired statistics used?

Regression analyses –

The regression model specification may have led to misleading results. Both external and internal bacterial concentration are very likely the result of the same, or highly correlated, hygienic practices. As such, external contamination may be considered as an exogenous variable. In a causal framework, external contamination may be thought of as on the causal pathway between hygiene behaviors and internal contamination. If external contamination is included in the model then the effect estimated for the other factors is only that effect that is independent of the relationship between the factor and external contamination. The authors should re-estimate their models leaving external contamination out of the specification in order to see the full effect of the behavior variables on internal contamination. This is the most significant limitation of this manuscript.

Line 206: A lack of statistical association between source water and internal/external water does not necessarily mean a lack of growth. Die-off will affect this relationship and net growth may be overwhelmed by contamination during processing.

Line 216: What is known about any treatment of treated sources? Are these perhaps just better described as ‘protected sources’ or ‘improved sources? The use of the word ‘treated’ may be misleading and potentially confused with treatment by the producer.

Line 238: cite the guidelines you are referring to.

Line 241-246: The point made here is accurate. However the bigger point is that the external contamination points to the processing process as the cause of the internal contamination. In sum, both the internal and external contamination are the result of the same unhygienic practices. This is covered in lines 252-254. See comment above regarding regression analyses.

Line 252: The evidence does not indicate that the external quality had an impact on internal quality; they exhibit a statistical association only.

The information about the water suppliers (line 260 and on) is very relevant to the discussion above. This information should be presented in the introduction or earlier in the Discussion.

The discussion should include commentary on passible interventions.

The negative association between use of a cup for abstraction (vs. a tap) on bacterial concentration should be discussed.

The results contain many sentences with interpretation of the results better suited for the discussion section.

Author Response

Thanks very much for your helpful comments-the detailed replies are attached

Reviewer 2 Report

This is a well written case study of water quality in sachets purchased in Malawi. Because the practice of sachet vending exists throughout the world, this article will have wide-spread interest. Slight improvements to the manuscript should be made before publication as follows:

Line 44:  WHO guideline is for E. coli not fecal coliform which is the test for thermotolerance.

Line 45: insert reference for the WHO guideline.

WHO | Guidelines for drinking-water quality, fourth edition. (n.d.). Retrieved March 17, 2019, from https://www.who.int/water_sanitation_health/publications/2011/dwq_guidelines/ en/

Line 46: The meaning of “sources” is unclear here. Generally “sources” are the river, lake or well. This seems to be “collected water to be used for consumption and hygiene”.

Line 69: Please clarify the units on your amount. Check to see if the jounal has a standard (USD or Euros) and use that.  For example  $0.07 USD.

Line 103:  Insert the word “sterile”. I assume from table 1 that that 100 mL was sterile distilled water but it’s important and should be noted.

Line 108:  It’s ok that the tap wasn’t disinfected or flamed, but it should flow so not to catch biofilm that accumulates on a wet tap. Did the protocol include allowing the tap to flow for x minutes before sample collection?  Protocols should have enough detail so that another team could repeat the work and would (hopefully) have the same result. Therefore, if water did not flow from the tap before collection, state that it was a “first draw” sample.

Line 139: This inference regarding education level can not be drawn from data presented in this manuscript. The data on educational level can be presented, but not the association to water vending, especially in regard to literacy. Easiest would be to just delete the highlighted sentence.

Line 145: According to the MDG and SDG definition of “improved” water sources, some groundwater sources are “improved”, so this sentence is not quite correct. Please remove the inference and simply state: All source water used for packaging and selling was purchased by the vendor. All source water was from groundwater.

Line 163: Indicate whether your are referencing the 1997, 2004 or 2011 guidelines. This appears to be the 1997 guideline (inclusion of TC in the “conformity” discussion).

Lines 227-230: “colonies” is not a unit. Change to CFU/mL or CFU/100mL.

Author Response

Thanks very much for your helpful comments and suggestions.  Detailed replies are attached.

Reviewer 3 Report

E. coli analysis

Introduction:

The introduction needs to be streamlined, it is quite disjointed, flipping between hand-tied and industrially produced and water quality of stored water..  It is recommended to begin with a description of sachet water and the SDGs/role in humanitarian context.  To follow with a summary of results from previous studies (in Tabular form, with data, not vague references), and a statement as to the novelty of this data (is this data novel?).  It appears "hand-tied" sachet water is novel.  However, it is quite unclear how hand-tied sachet water is used in humanitarian contexts.   Additional information is needed as to the relevance in humanitarian contexts.

39 - not always 500 mL

45 - check standard (0 or <1)?

46 - the relevance of the Wright study is unclear

Figure 1.  Unnecessary and does not add value. Remove. 

Methods

Was the distilled water used to collect the exterior E. coli sample buffered in any way?  What was the recovery method from surfaces of this method?  What is the reference for the use of this method?

E. coli should be italicized throughout

Consistency - ml or mL throughout 

Results

It is not appropriate to analyze E. coli surface data using E. coli water sample methods.  Surface data should be reported in CFU/cm2 of surface.  And standards for E. coli on surfaces should be used, not water standards. 

Table 4:

p-values of 0.000 are not mathematically possible

Regressions

There needs to be more thought put into comparing E. coli results in water versus E. coli results on surfaces.  They are not comparable directly as stated in E. coli result tables or regressions.  The recovery rate of E. coli off the surface using the method described is not presented in the manuscript, and the risk from ingested water versus surface water will be different.  More thought needs to be completed as to risk differences - this is a complex statement - is this accurate?  Where is the supporting data for this?    "The consumer places her mouth over  the corner of the sachet, rips off a piece of the plastic and drains the liquid by sucking it out. In the  process, her mouth is in direct contact with the external surface and, as she drinks, is passing the  internal liquid over the external surface, thereby transferring additional contamination into the liquid consumed."   I have seen users ripping the sachet open and then pouring water into their mouths with no contact (it's hard to suck on a flexible bag)... the risk of the external surface versus internal water needs more analysis and thought. 

Discussion

280 - what is a "weak" chlorine solution?  Please be more specific

Additional

The manuscript would be improved by images of the sachets

Author Response

(The authors gave the same response as above.)

Round 2

Reviewer 3 Report

Thank you for addressing the points.  The only remaining comment I have I think recovery of E. coli from hands is clearly correlated to recovery of E. coli from surfaces.  There is significant literature on E. coli on surfaces and recovery rates that could be referenced. 

Author Response

We have added two additional references that we are familiar with, but would be happy to include additional material if the Reviewer thinks we have missed something important.